# Higher Risk of Tumor Recurrence in NASH-Related Hepatocellular Carcinoma Following Curative Resection

**DOI:** 10.3390/v14112427

**Published:** 2022-10-31

**Authors:** Shih-Chieh Chien, Yih-Jyh Lin, Chun-Te Lee, Yen-Cheng Chiu, Tsung-Ching Chou, Hung-Chih Chiu, Hung-Wen Tsai, Che-Min Su, Tsung-Han Yang, Hsueh-Chien Chiang, Wei-Chu Tsai, Kai-Chun Yang, Pin-Nan Cheng

**Affiliations:** 1Department of Internal Medicine, National Cheng Kung University Hospital, College of Medicine, National Cheng Kung University, Tainan 701, Taiwan; 2Department of Surgery, National Cheng Kung University Hospital, College of Medicine, National Cheng Kung University, Tainan 701, Taiwan; 3Department of Pathology, National Cheng Kung University Hospital, College of Medicine, National Cheng Kung University, Tainan 701, Taiwan

**Keywords:** non-alcoholic steatohepatitis, hepatocellular carcinoma, curative resection

## Abstract

**Background:** The outcomes for patients with NASH-related HCC after curative resection have not been clarified. This study compared the overall survival (OS), time-to-tumor recurrence (TTR), and recurrence-free survival (RFS) associated with NASH-related HCC and virus-related HCC after resection. **Methods:** Patients with HCC who underwent curative resection were retrospectively enrolled. Baseline characteristics, including disease etiologies and clinical and tumor features, were reviewed. The primary outcomes were OS, TTR, and RFS. **Results:** Two hundred and six patients were enrolled (HBV: *n* = 121, HCV: *n* = 54, NASH: *n* = 31). Of those with virus-related HCC, 84.0% achieved viral suppression. In both the overall and propensity-score-matched cohorts, those with NASH-related HCC experienced recurrence significantly earlier than those with virus-related HCC (median TTR: 1108 days vs. non-reached; *p* = 0.03). Through multivariate analysis, NASH-related HCC (hazard ratio (HR), 2.27; 95% confidence interval (CI), 1.25–4.12) was independently associated with early recurrence. The unadjusted RFS rate of the NASH-related HCC group was lower than the virus-related HCC group. There was no difference in the OS between the two groups. **Conclusions:** NASH-related HCC was associated with earlier tumor recurrence following curative resection compared to virus-related HCC. Post-surgical surveillance is crucial for detecting early recurrence in patients with NASH-related HCC.

## 1. Introduction

Non-alcoholic steatohepatitis (NASH) is an emerging global disease, especially in the Asia-Pacific region, [1] and it is becoming an increasingly prominent cause of hepatocellular carcinoma (HCC), liver transplantation, and liver-related death [2,3,4,5,6]. In contrast to viral hepatitis, NASH-related HCC is frequently diagnosed late because of a lack of adequate surveillance strategies and effective treatments for halting liver disease progression [5,7]. In addition, the risk factors for NASH (e.g., physical inactivity, insulin resistance, and lipotoxicity) continuously cause liver injury, leading to increased disease progression or even HCC development and growth [8,9,10]. Studies have compared NASH-related HCC with hepatitis B virus (HBV)- and hepatitis C virus (HCV)-related HCC, and they have generally reported more favorable disease-free and overall survival (OS) for NASH-related HCC [5,7,11,12,13,14,15] and less favorable outcomes for patients with concurrent NASH and HBV relative to patients with only HBV-related HCC [16]. The advent of effective antiviral treatments has improved the outcomes of virus-related HCC to the extent that such improvements are expected and observable [17,18,19]. However, the effects of NASH are irreversible because a lack of effective disease-specific treatments [20,21] can negatively influence HCC-associated outcomes. Therefore, the outcomes of HCCs with different etiologies (e.g., NASH- and virus-related HCCs) should be further evaluated.

The present retrospective study compared the outcomes of patients with NASH- and virus-related HCC after curative resection. The factors that could influence their outcomes (e.g., patient-related and tumoral factors) were also analyzed.

## 2. Materials and Methods

### 2.1. Patient Selection and Outcome Measurement

The data of patients with newly diagnosed HCC who underwent initial curative surgical resection between January 2016 and December 2019 were retrospectively collected from the database of National Cheng Kung University Hospital, a tertiary medical center in Taiwan. The date of the end of follow-up was 30 June 2022. The primary outcomes were OS and time-to-tumor recurrence (TTR), which were defined as the time from diagnosis to death and the time to first HCC recurrence following initial HCC resection, respectively. The 2022 updated version of the Barcelona clinic liver cancer classification (BCLC) system was used to stage HCC [22]. Curative surgical resection was defined as the achievement of a clear microscopic margin without tumor involvement and the complete resection of a tumor (R0 resection). HCC recurrence was diagnosed by evaluating contrast-enhanced dynamic images, including computed tomography and magnetic resonance images, in accordance with standard imaging criteria [23].

### 2.2. Diagnosis of Liver Diseases

The diagnosis of HBV infection was confirmed through a positive serum HBV surface antigen (HBsAg) test, the presence of HBV core antigen, the presence of HBsAg under immunohistological staining, or a combination of the three; HCV infection was confirmed by a positive serum anti-HCV antibody test. The diagnosis of NASH was established through a histopathological examination of the non-tumor part of liver tissue, which was evaluated by a pathologist from the hospital (Dr. Hung-Wen Tsai) who specialized in liver histology. NASH severity was graded using the Brunt system [24,25].

### 2.3. Clinical and Histopathological Characteristics

Through a thorough review of medical records, the clinical features of the included patients, including age, sex, body mass index (BMI), biochemical and hematological test results, and alpha-fetoprotein (AFP) concentrations, were obtained and recorded. Patients with chronic hepatitis B who were treated by an antiviral agent and had an undetectable or low titer of HBV DNA (<20 IU/mL) before curative resection or patients with chronic hepatitis C who achieved a sustained viral response before curative resection were regarded as having achieved viral suppression. All histopathological characteristics, including the fibrosis stage, as evaluated using the METAVIR staging system; level of tumor cell differentiation; level of tumor capsulation; status of capsular invasion; and presence or absence of microvascular invasion and satellite nodules, were recorded or graded.

Patients were excluded if they had BCLC stage C or D, sarcomatoid HCC, or coexisted with intrahepatic cholangiocarcinoma or hepatocholangiocarcinoma, two coexisting etiologies of liver disease (e.g., HBV and HCV coinfection or viral hepatitis plus NASH), autoimmune hepatitis, primary biliary cholangitis, Wilson disease, or hemochromatosis.

### 2.4. Statistical Analysis

The baseline characteristics of multiple groups (HBV, HCV, and NASH) were compared. Continuous variables were analyzed through an analysis of variance (ANOVA) with corresponding subgroup analyses, and categorical variables were analyzed through Chi-square tests. The time-dependent outcomes were calculated and graphed by the Kaplan–Meier survival method, and time differences among the groups, including OS, RFS, and TTR, were analyzed through log-rank tests. Cox regression analysis was conducted to determine the hazard ratios (HRs) and 95% confidence intervals (CIs) of the potential influencing factors for OS, RFS, and TTR. Any factor with a *p* value of <0.15 was included in a multivariate analysis conducted using backward elimination [26,27]. The number of factors included in the multivariate analysis was determined by the number of events [28]. For factors with potential multicollinearity (determined by a variance inflation factor, VIF, equal to or greater than 4), only one of the factors was selected for the multivariate analysis. Propensity-score matching was applied to match the NASH-related HCC and HBV-related HCC subgroups based on the results obtained before matching. The one-to-one nearest neighbor matching method (without replacements) was applied and the caliper was set to 0.2 [29]. All the results with a *p* value of <0.05 were considered significant.

## 3. Results

### 3.1. Baseline Characteristics

Figure 1 presents the algorithm for patient selection. In total, 206 patients (121 with HBV-related HCC, 54 with HCV-related HCC, and 31 with NASH-related HCC) were enrolled. Table 1 lists the baseline characteristics of the enrolled patients. Compared with the virus-related HCC groups, the NASH-related HCC group had a significantly higher prevalence of type II diabetes mellitus and hypertension and was significantly heavier. The AFP concentrations, Child–Pugh scores, albumin–bilirubin (ALBI) scores, and aspartate aminotransferase (AST) to platelet ratio index (APRI) scores of the three groups were comparable.

A mean steatosis level of 25.6% was present in the non-tumor parts of the liver tissues obtained from the patients with NASH-related HCC. The median necroinflammatory grade (ranging from 1 to 2) of NASH was grade 1. None of the enrolled patients had macrovascular invasion and 200 (97.1%) were classified as having BCLC stage 0–A. The HCV-related HCC group had a higher proportion of well-capsulated tumors relative to the HBV- and NASH-related HCC groups. The other HCC characteristics, including the tumor size, tumor number, level of tumor cell differentiation, presence of satellite nodules, and presence of microvascular invasion, were similar among the three groups.

### 3.2. OS Analysis

During the study period, 24 patients died from tumor-related causes or cirrhosis-related comorbidities. The 1-year, 3-year, and 5-year OS rates were 97.1%, 91.2%, and 88.3%, respectively. Through multivariate analysis, tumor recurrence (HR, 4.43; 95% CI, 1.72–11.42; *p* = 0.002), BCLC stage B (HR, 12.57; 95% CI, 2.61–60.52; *p* = 0.002), and absence of tumor capsulation (HR, 5.88; 95% CI, 1.59–21.70; *p* = 0.01) were revealed to be significantly associated with an increased risk of death (Table 2). No significant difference in the OS rates was identified between the virus-related HCC and NASH-related HCC groups (Figure 2a,b).

### 3.3. Recurrence of HCC after Curative Resection

For the overall cohort, the cumulative incidences of 1-year, 3-year, and 5-year tumor recurrence were 14.6%, 31.7%, and 37.6%, respectively. The NASH-related HCC group exhibited a significantly shorter median TTR relative to the virus-related HCC group (1108 days vs. non-reached; log-rank *p* = 0.03; Figure 2c,d). This was particularly true for the HBV-related HCC group. The 5-year tumor recurrence rate of the NASH-HCC group was significantly higher than that of the HBV-HCC group (NASH-HCC vs. HBV-HCC, 48.4% vs. 33.9%; *p* = 0.002) but not that of the HCV-related HCC group (NASH-HCC vs. HCV-related HCC, 48.4% vs. 38.9%; *p* = 0.2). Multivariate analysis revealed that NASH-related HCC (HR, 2.27; 95% CI, 1.25–4.12; *p* = 0.01), the presence of satellite nodules (HR, 1.92; 95% CI, 1.01–3.65; *p* = 0.045), Child–Pugh stage B (HR, 5.17; 95%CI, 1.58–16.95; *p* = 0.01), and a pre-surgical AFP concentration of >20 ng/mL (HR, 1.72; 95% CI, 1.05–2.83) were associated with early recurrence. No significant association was identified for the other investigated factors, namely the presence of cirrhosis, microvascular invasion, invasion of the tumor capsule by malignant cells, poor differentiation of tumor cells, and baseline liver function (Table 2).

The 1-year, 3-year, and 5-year recurrence-free survival (RFS) rates of the overall cohort were 84.0%, 66.5%, and 59.2%, respectively. The NASH-related HCC group exhibited a significantly shorter RFS relative to the virus-related HCC group, and the log-rank test based on the Kaplan–Meier survival assumption showed a *p* = 0.037 (Figure 2e,f). However, through multivariate regression analysis, the adjusted RFS rate of the NASH-related HCC group was revealed to be similar to that of the virus-related HCC group (the adjusted HR was 1.51, 95% CI:0.81-2.81, Table 2).

### 3.4. Effect of Antiviral Treatment on Outcomes

We further analyzed the effect of viral suppression status on OS and TTR. Overall, 137 patients (84.0%) with virus-related HCC underwent antiviral treatment. Among them, 54 (33.1%) and 83 (50.9%) patients were treated before and after curative resection, respectively. Thirty-one patients with HBV-related HCC (34.8%) and 16 with HCV-related HCC (34.8%) achieved viral suppression before curative resection. Non-significant trends toward improved OS and TTR were identified in patients with viral suppression before curative resection (log-rank *p* = 0.28 and 0.35 for OS and TTR, respectively; Figure 2g,h).

### 3.5. Propensity-Score-Matching Analysis of NASH-Related and HBV-Related HCC Groups

The results of the present study reveal that the NASH-related HCC group had significantly poorer TTR and RFS relative to the HBV-related HCC group (Figure 2c,e). We subsequently matched these two groups through propensity-score matching. The factors that could influence tumor recurrence were matched; they comprised age, sex, BMI, presence of diabetes, maximum tumor length, tumor number, level of tumor cell differentiation, tumor capsulation and tumor capsule invasiveness, the presence of satellite nodules, microvascular invasion, the presence of cirrhosis, baseline liver function, ALBI score, and baseline serum AFP level. Through matching, the data of 52 patients with an equal number of NASH-related (*n* = 26) and HBV-related HCC (*n* = 26) were included for further analysis. The baseline characteristics of the two groups were comparable (Table 3). Kaplan–Meier analysis revealed that the patients with NASH-related HCC had significantly earlier recurrence relative to those with HBV-related HCC (log-rank *p* = 0.03) and that the OS (*p* = 0.19) and RFS (*p* = 0.06) of the two groups were non-significantly different (Figure 3).

## 4. Discussion

In the present study, the outcomes of patients with NASH-related HCC after surgical resection and the identified risk factors for tumor recurrence were similar to those reported by other studies [5,18,30,31,32,33]. The reported risk factors for tumor recurrence are the presence of satellite nodules, high baseline AFP levels, poor liver function, and the presence of cirrhosis. The outcomes of the patients from the three etiological groups of the present study were compared, and significantly shorter TTR and lower RFS were identified in patients with NASH-related HCC relative to those with virus-related HCC, especially when compared with HBV-related HCC. These findings were validated through propensity-score matching performed to control for potential confounders of HCC recurrence between patients with HBV-related and NASH-related HCC. These results are different from those of most other studies, which have reported that patients with NASH-related HCC had more favorable or comparable OS and recurrence rates relative to patients with virus-related HCC [5,11,12,13,14,15,34,35].

In the present study, the patients with NASH-related HCC had a 5-year TTR rate of 48.8%, which is comparable to the rates reported in other studies (36–76%) [34,35]. In addition, the 5-year RFS rate of 48.4% obtained in the present study was similar to the 34–87% reported by other studies [5,12,15,34,35]. By contrast, the 5-year RFS rate of 63.6% of the virus-related HCC group of the present study was more favorable than the results obtained in other studies (30–39%) [13,34,35]. We suggest that the viral suppression status may be a key reason for this finding. In the present study, 84.0% of enrolled patients with virus-related HCC underwent antiviral therapy and achieved viral suppression before or after surgery; this is higher than the reported 0–23.4% of patients in other studies, which discuss similar issues [12,13,34,35]. As mentioned above, effective treatments for regressing or improving the long-term outcomes of NASH are lacking. Furthermore, compared with patients with virus-related HCC, patients with NASH-related HCC were more likely to be overweight (83.6%, 59.3%, and 62.0% of patients with NASH-related, HBV-related, and HCV-related HCC, respectively, had a BMI of ≥23; *p* = 0.048), have type II diabetes mellitus (DM; NASH-related HCC vs. HBV-related and HCV-related HCC, 61.3% vs. 38.9% and 17.4%; *p* < 0.001), have hypertension (NASH-related HCC vs. HBV-related and HCV-related HCC; 77.4% vs. 64.8% and 45.5%; *p* = 0.002), and have severe stages of fibrosis (*p* = 0.04). These aforementioned factors could have contributed to the higher tumor recurrence rate in the NASH-related HCC group. Studies have indicated that having type II DM, metabolic comorbidities (e.g., overweight and hypertension), or a combination of both increases the risk of HCC development [9,36,37,38]. Several mechanisms have been proposed to explain HCC development in patients with DM and metabolic syndromes. They include carcinogenesis promoted by the activation of insulin growth factors [39] and the presence of a chronic proinflammatory microenvironment that causes repetitive hepatocellular cytotoxicity and genomic instability [36,39,40]. NASH-related aberrant T-cell activation also results in an immunosuppressive status that may further impair immune surveillance [40,41].

The outcomes of HBV-related HCC in the present study were different from those reported by other studies. Yang et al. applied propensity-score matching to analyze the outcomes of NASH-related and HBV-related HCC groups and revealed that the OS and RFS rates of these two groups were similar. In their study, a greater number of patients with NASH-related HCC experienced macrovascular HCC invasion (10.4% and 14.1% of patients with HBV-related and NASH-related HCC, respectively), multiple tumor nodules (25.0% and 26.4% of patients with HBV-related and NASH-related HCC, respectively), microvascular invasion (53.1% and 55.7% of patients with HBV-related and NASH-related HCC, respectively), satellite nodules (28.1% and 28.3% of patients with HBV-related and NASH-related HCC, respectively), and mostly poorly differentiated tumor cells (72.9% and 82.4% of patients with HBV-related and NASH-related HCC, respectively). Yang et al. also reported a lower 5-year survival rate (NAFLD vs. HBV, 46.3% vs. 43.5%) and higher recurrence rate (NAFLD vs. HBV, 64.6% vs. 71.5%) among patients with HBV relative to those with NAFLD [13]. By contrast, all patients in the present study had no macrovascular invasion; most had a solitary tumor (HBV-related HCC vs. NASH-related HCC, 92.6% vs. 87.1%), no microvascular invasion (HBV-related HCC vs. NASH-related HCC, 75.2% vs. 77.4%), no satellite nodules (HBV-related HCC vs. NASH-related HCC, 86.0% vs. 96.8%), and mostly well to moderately differentiated tumor cells (HBV-related HCC vs. NASH-related HCC, 91.7% vs. 93.5%). Compared with other studies, the present study reported higher OS and a lower TTR. Notably, 82% of the patients with HBV-related HCC in the present study underwent antiviral therapy and achieved viral suppression, which contributed to the beneficial impact on the secondary prevention of HCC outcomes [42,43,44], especially the prevention of late recurrence [42]. By contrast, a lack of effective treatment, risk stratification, and adequate surveillance strategies for patients with NASH negatively influence the outcomes following HCC curative resection and therefore result in differing outcomes in patients with NASH-related and HBV-related HCC. In the present study, propensity-score matching was performed to control for potential confounders of HCC outcomes, and the results consistently indicated a significantly shorter TTR in the NASH-related HCC group relative to the other groups, which validated our discovery.

The present study has several limitations. First, the retrospective design may exhibit bias. Second, the sample size of patients with NASH-related HCC was small. However, a significant difference in the TTR was still identified between the NASH-related HCC and virus-related HCC groups. Third, the HCC stage at recurrence and subsequent treatment, which may influence OS, were not included in the analysis.

## 5. Conclusions

In summary, in a modern healthcare environment with highly effective antiviral treatments, the present study revealed a shorter TTR following curative HCC resection in patients with NASH-related HCC relative to those with virus-related HCC, especially HBV-related HCC. In contrast to HBV or HCV, effective treatment for NASH is currently lacking and is urgently required, not only to halt disease progression but also to prevent the recurrence of NASH-related HCC. Physicians treating patients with NASH-related HCC should focus on strategies for post-surgical surveillance.

## Figures and Tables

**Figure 1 viruses-14-02427-f001:**
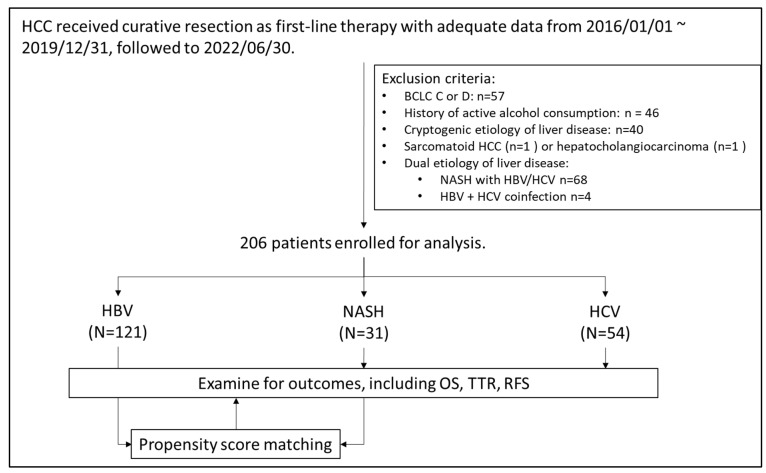
Algorism of patient selection.

**Figure 2 viruses-14-02427-f002:**
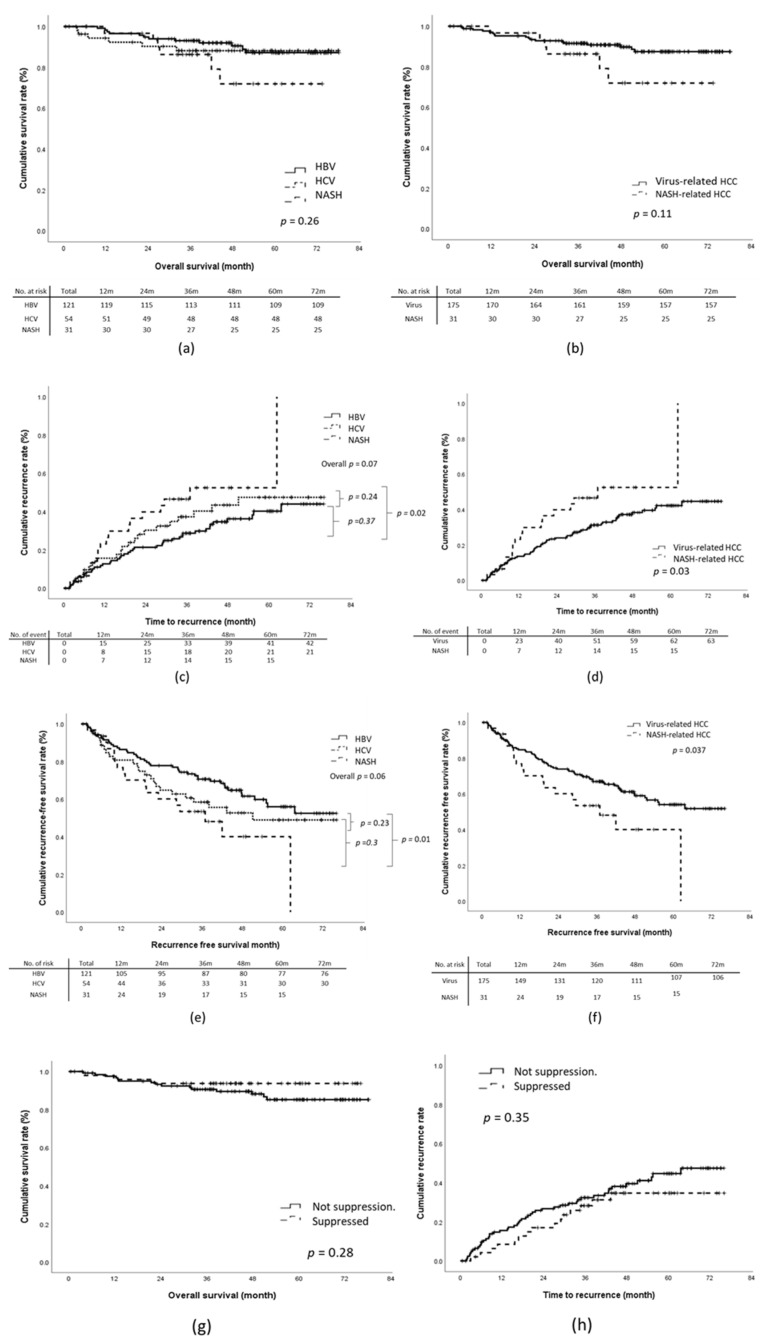
Outcomes according to etiology of liver disease: (**a**,**b**) overall survival (OS), (**c**,**d**) time-to-recurrence (TTR), (**e**,**f**) recurrence-free survival (RFS). (**g**,**h**): OS and TTR according to viral suppression status.

**Figure 3 viruses-14-02427-f003:**
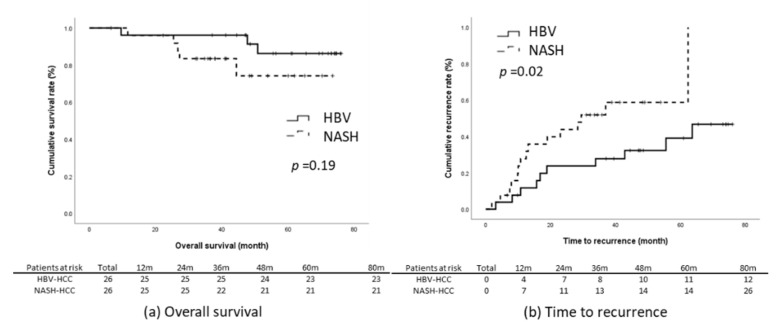
Outcomes in the propensity-score-matched cohort. (**a**) OS, (**b**) TTR, (**c**) RFS.

**Table 1 viruses-14-02427-t001:** Baseline characteristics of patients according to etiology of liver disease.

		HBV (*n* = 121)	HCV (*n* = 54)	NASH (*n* = 31)	*p* Value
Age		60 (32–88)	68 (48–83)	66 (52–80)	0.000
Gender	Male vs. Female	96 (79.3%) vs. 25 (20.7%)	36 (66.7%) vs. 18 (33.3%)	23 (74.2%) vs. 8 (25.8%)	0.20
Body weight (kg)		65.0 (33.7–110.0)	60.2 (38.8–95.5)	70.0 (40.6–87.0)	0.10
BMI (kg/m^2^)		24.1 (14.4–34.3)	23.9 (18.5–32.8)	26.2 (18.0–31.9)	0.004
Type II diabetes	Absent vs. Present	100 (82.6%) vs. 21 (17.4%)	33 (61.1%) vs. 21 (38.9%)	12 (38.7%) vs. 19 (61.3%)	<0.001
Hypertension	Absent vs. Present	66 (54.5%)/55 (45.5%)	19 (35.2%)/35 (64.8%)	7 (22.6%)/24 (77.4%)	0.002
Hyperlipidemia	Absent vs. Present	98 (81.0%)/23 (19.0%)	40 (74.1%)/14 (25.9%)	22 (71.0%)/9 (29.0%)	0.370
BCLC stage	0/Solitary/Multinodular/B	29 (24.0%)/83 (68.6%)/6 (5.0%)/3 (2.5%)	14 (25.9%)/36 (66.7%)/2 (3.7%)/2 (3.7%)	5 (16.1%)/22 (71.0%)/3 (9.7%)/1 (3.2%)	0.87
Maximal Tumor length (cm)		3.1 (0.9–26.0)	2.8 (0.8–11.5)	3.5 (0.9–8.0)	0.17
Tumor number	1 vs. 2	112 (92.6%)/9 (7.4%)	50 (92.6%)/4 (7.4%)	27 (87.1%)/4 (12.9%)	0.59
Histologic differentiation	Well/moderate/poor	26 (21.5%)/85 (70.2%)/10 (8.3%)	10 (18.5%)/35 (64.8%)/9 (16.7%)	4 (12.9%)/25 (80.6%)/2 (6.5%)	0.31
Cytological grade	G1/G2/G3	19 (16.7%)/66 (57.9%)/29 (25.4%)	9 (17.0%)/24 (45.3%)/20 (37.7%)	4 (13.3%)/21 (70.0%)/5 (16.7%)	0.21
Tumor capsulation	Well capsulated	27 (22.3%)	23 (42.6%)	7 (22.6%)	0.03
	Partly capsulated	70 (57.9%)	18 (33.3%)	16 (51.6%)	
	Non-capsulated	24 (19.8%)	13 (24.1%)	8 (25.8%)	
Invasion of tumor capsule	No vs. Yes	62 (51.2%) vs. 59 (48.8%)	29 (53.7%) vs. 25 (46.3%)	22 (71.0%) vs. 9 (29.0%)	0.14
Satellite nodule	Absent vs. presented	104 (86.0%) vs. 17 (14.0%)	50 (92.6%) vs. 4 (7.4%)	30 (96.8%) vs. 1 (3.2%)	0.15
Distance from margin	>10 mm vs. <10 mm	36 (30.0%) vs. 84 (70.0%)	13 (24.5%) vs. 40 (75.5%)	11 (35.5%) vs. 20 (64.5%)	0.56
Microvascular invasion	Absent vs. Presented	91 (75.2%) vs. 30 (24.8%)	40 (74.1%) vs. 14 (25.9%)	24 (77.4%) vs. 7 (22.6%)	0.94
Type of liver resection	Non-anatomical vs. anatomical	40 (33.1%)/88 (66.9%)	29 (53.7%)/25 (46.3%)	14 (45.2%)/17 (54.8%)	0.03
Fibrosis stage (Metavir)	½/3/4	20 (16.5%)/11 (9.1%)/46 (38.0%)/44 (36.4%)	1 (1.9%)/6 (11.1%)/26 (48.1%)/21 (38.9%)	4 (12.9%)/4 (12.9%)/6 (19.4%)/17 (54.8%)	0.04
Steatosis (%)		3.0% (0.0–35.0%)	2.5% (0.9–30.0%)	25.0% (5.0–65.0%)	0.000
Brunt inflammatory grade	No	121 (100%)	54 (100%)	0	NA.
	Grade 1	0	0	24 (77.4%)	
	Grade 2	0	0	7 (22.6%)	
Antiviral treatment	Before diagnosis	35 (31.5%)	19 (36.5%)	NA	0.26
	After diagnosis	56 (50.5%)	27 (51.9%)		
	No treatment and low viral load *	12 (10.8%)	1 (1.9%)		
	No treatment and high viral load *	8 (7.2%)	5 (9.6%)		
Viral suppression #	No suppression before surgery	58 (65.2%)	30 (65.2%)	NA.	0.58
	Suppressed before surgery ^$^	31 (34.8%)	16 (34.8%)		
Pre-surgical AFP (ng/mL)		10.6 (0.6–60,500.0)	5.5 (1.5–3519.0)	4.3 (0.9–3076.0)	0.43
ALT (U/L)		26 (10–162)	25 (9–258)	37 (10–180)	0.05
AST (U/L)		34 (14–205)	39 (16–140)	36 (16–93)	0.75
Platelet (10^3^/μL)		179 (75–420)	165 (66–446)	196 (75–311)	0.16
Alb (g/dL)		4.5 (3.0–5.4)	4.4 (1.8–5.3)	4.5 (3.5–5.3)	0.24
Bil-T (mg/dL)		0.6 (0.2–6.4)	0.6 (0.2–1.2)	0.6 (0.2–2.7)	0.41
PT (INR)		1.04 (0.88–1.27)	1.03 (0.88–1.24)	1.04 (0.90–1.14)	0.28
Ascites	Absent vs. presented	113 (95.0%) vs. 6 (5.0%)	48 (96.0%) vs. 2 (4.0%)	28 (93.3%) vs. 2 (6.7%)	0.87
Hepatic encephalopathy	No vs. presented.	120 (100.0%) vs. 0%	54 (100.0%) vs. 0%	31 (100.0%) vs. 0%	
CTP score	5/6/7	100 (92.6%)/4 (3.7%)/4 (3.7%)	47 (95.9%)/2 (4.1%)/0 (0.0%)	25 (89.3%)/3 (10.7%)/0 (0.0%)	0.26
APRI score		0.52 (0.11–1.93)	0.62 (0.11–3.02)	0.48 (0.17–1.92)	0.06
ALBI score		−3.20 (−3.88~−1.54)	−3.07 (−3.76~−2.48)	−3.07 (−3.95~−1.88)	0.76

* HBV DNA <1000 IU/mL, HCV RNA-positive. # Undetectable HBV DNA and/or undetectable HCV RNA. $ Spontaneously or through an antiviral drug.

**Table 2 viruses-14-02427-t002:** Univariate and multivariate analysis for outcomes, including overall survival (OS), time-to-recurrence (TTR), and recurrence-free survival (RFS).

		OS				TTR				RFS			
		Univariate Analysis		Multivariate Analysis		Univariate Analysis		Multivariate Analysis		Univariate Analysis		Multivariate Analysis	
		HR (95% CI)	*p* Value	HR (95% CI)	*p* Value	HR (95% CI)	*p* Value	HR (95% CI)	*p* Value	HR (95% CI)	*p* Value	HR (95% CI)	*p* Value
Etiology	NASH vs. viral	2.09 (0.83–5.29)	0.12	1.81 (0.63–5.21)	0.27	1.81 (1.04–3.15)	0.036	2.27 (1.25–4.12)	0.01	1.75 (1.03–3.00)	0.040	1.51 (0.81–2.81)	0.20
Age	>60 vs. ≤60	1.20 (0.52–2.81)	0.67			1.03 (0.65–1.63)	0.90			1.06 (0.68–1.65)	0.79		
Gender	Female vs. Male	1.23 (0.51–2.97)	0.64			0.95 (0.56–1.59)	0.83			1.02 (0.63–1.66)	0.94		
BMI	<23	Ref.	0.96			Ref.	0.44			Ref.	0.43		
	23–30	1.09 (0.46–2.56)	0.85			1.33 (0.82–2.17)	0.25			1.34 (0.84–2.14)	0.22		
	>30	0.83 (0.10–6.66)	0.86			1.59 (0.60–4.16)	0.35			1.46 (0.56–3.80)	0.44		
Type II DM	Presented vs. Absent	1.65 (0.73–3.72)	0.22			1.20 (0.75–1.93)	0.44			1.46 (0.94–2.26)	0.09	1.16 (0.69–1.95)	0.57
Hypertension	Presented vs. Absent	1.63 (0.70–3.81)	0.26			1.26 (0.80–1.98)	0.32			1.30 (0.84–2.01)	0.23		
Hyperlipidemia	Presented vs. Absent	1.39 (0.58–3.36)	0.46			0.95 (0.55–1.62)	0.84			1.05 (0.64–1.73)	0.85		
BCLC stage	0	Ref.	0.03	Ref.	0.01	Ref.	0.99			Ref.	0.91		
	Solitary	1.49 (0.50–4.47)	0.47	2.06 (0.67–6.35)	0.21	1.08 (0.64–1.82)	0.77			1.05 (0.64–1.72)	0.86		
	Multiple within Milan	0.99 (0.11–8.82)	0.99	0.56 (0.06–5.06)	0.60	1.01 (0.34–2.96)	0.99			0.91 (0.31–2.64)	0.86		
	B	7.97 (1.78–35.69)	0.01	12.57 (2.61–60.52)	0.0016	1.15 (0.27–4.96)	0.85			1.53 (0.46–5.13)	0.49		
Tumor number	2 vs. 1	2.12 (0.72–6.20)	0.17			0.99 (0.43–2.28)	0.99			1.06 (0.49–2.30)	0.88		
Tumor length (cm)	<2	Ref.	0.75			Ref.	0.88			Ref.	0.99		
	2–4.9	1.37 (0.49–3.80)	0.55			1.01 (0.60–1.70)	0.97			1.03 (0.62–1.69)	0.92		
	≥5	1.59 (0.46–5.50)	0.46			1.17 (0.60–2.26)	0.65			1.05 (0.55–2.02)	0.88		
Histologic differentiation	Well	Ref.	0.60			Ref.	0.48				0.47		
	Moderate	1.25 (0.42–3.67)	0.69			1.40 (0.77–2.56)	0.27			1.33 (0.76–2.33)	0.33		
	Poor	0.46 (0.05–4.15)	0.49			1.07 (0.43–2.69)	0.88			0.94 (0.38–2.30)	0.89		
Cytological grade	Grade 1	Ref.	0.92			Ref.	0.44				0.59		
	Grade 2	0.85 (0.27–2.64)	0.78			1.55 (0.76–3.19)	0.23			1.37 (0.71–2.65)	0.35		
	Grades 3–4	1.02 (0.30–3.47)	0.98			1.61 (0.74–3.49)	0.23			1.43 (0.70–2.93)	0.32		
Tumor Capsulation	Well	Ref.	0.00	Ref.	0.001	Ref.	0.89				0.54		
	Partial	1.52 (0.41–5.62)	0.53	1.30 (0.35–4.85)	0.70	1.07 (0.63–1.83)	0.80			1.00 (0.60–1.68)	0.99		
	No capsule	5.28 (1.49–18.71)	0.01	5.88 (1.59–21.70)	0.01	1.17 (0.62–2.21)	0.63			1.32 (0.73–2.38)	0.36		
Invasion of tumor capsule	Yes vs. no	1.39 (0.62–3.11)	0.42			1.42 (0.91–2.21)	0.12	1.31 (0.81–2.09)	0.27	1.31 (0.86–2.00)	0.21		
Satellite nodule	Present vs. absent.	2.88 (1.14–7.26)	0.03			2.02 (1.11–3.67)	0.02	1.92 (1.01–3.65)	0.045	1.84 (1.02–3.32)	0.04	1.66 (0.84–3.27)	0.14
Safety margin (mm)	≥10 vs. <10	1.55 (0.58–4.16)	0.38			0.97 (0.59–1.57)	0.89			0.98 (0.61–1.56)	0.93		
Microvascular invasion	Present vs. absent	1.50 (0.64–3.51)	0.35			1.55 (0.96–2.49)	0.07	1.20 (0.69–2.08)	0.52	1.44 (0.91–2.28)	0.12	1.38 (0.81–2.34)	0.24
Resection type	Anatomical vs. non-anatomical resection	0.44 (0.19–0.98)	0.05	0.58 (0.23–1.46)	0.25	0.62 (0.40–0.97)	0.04	0.65 (0.41–1.03)	0.07	0.60 (0.39–0.92)	0.02	0.65 (0.42–1.01)	0.055
Recurrence after surgery	Yes vs. no	5.10 (1.88–13.83)	0.00	4.43 (1.72–11.42)	0.002								
Cirrhosis	Present vs. Absent	1.14 (0.51–2.56)	0.75			1.92 (1.23–2.99)	0.00	1.41 (0.88–2.26)	0.16	1.96 (1.28–3.01)	0.00	1.76 (1.14–2.73)	0.01
CTP score	A5	Ref.	0.47	Ref.	0.95	Ref.	0.08	(-)	0.02		0.05	(-)	0.06
	A6	1.94 (0.45–8.28)	0.37	1.31 (0.26–6.73)	0.75	1.26 (0.46–3.45)	0.65	1.14 (0.39–3.32)	0.82	1.76 (0.77–4.06)	0.18	1.30 (0.52–3.24)	0.57
	B7	2.51 (0.34–18.80)	0.37	0.92 (0.08–10.44)	0.95	3.68 (1.15–11.78)	0.03	5.17 (1.58–16.95)	0.01	3.50 (1.10–11.19)	0.03	4.09 (1.25–13.37)	0.02
APRI	APRI >1 vs. <1	1.20 (0.41–3.50)	0.74			1.68 (0.96–2.95)	0.07			1.64 (0.95–2.82)	0.08		
ALBI grade	Grade 2 vs. Grade 1	2.43 (0.83–7.15)	0.11	1.86 (0.44–7.90)	0.40	1.28 (0.59–2.78)	0.54			1.33 (0.64–2.76)	0.44		
Baseline AFP	>20 vs. ≤20	1.09 (0.46–2.54)	0.85			1.49 (0.95–2.33)	0.08	1.72 (1.05–2.83)	0.03	1.36 (0.88–2.09)	0.17		

**Table 3 viruses-14-02427-t003:** Baseline characteristics of patients with HBV- and NASH-related HCC after propensity-score matching.

		HBV (*n* = 26)	NASH (*n* = 26)	
		Median (Range) or *n* (%)	Median (Range) or *n* (%)	*p* Value
Age		65 (37–77)	66 (52–80)	0.14
Gender	Male vs. female	20 (76.9%) vs. 6 (23.1%)	20 (76.9%) vs. 6 (23.1%)	1.00
BMI		25.9 (19.6–33.0)	26.3 (22.1–31.9)	0.45
Type II diabetes	Absent vs. present	16 (61.5%) vs. 10 (38.5%)	15 (57.7%) vs. 11 (42.3%)	0.78
BCLC stage	0/solitary/multinodular within Milan/B	4 (15.4%)/20 (76.9%)/1 (3.8%)/1 (3.8%)	4 (15.4%)/19 (73.1%)/2 (7.7%)/1 (3.8%)	0.95
Maximal Tumor length (cm)		3.5 (1.2–14.0)	4.0 (0.9–8.0)	1.00
Tumor number	1 vs. 2	24 (92.3%) vs. 2 (7.7%)	23 (88.5%) vs. 3 (11.5%)	0.64
Histologic differentiation	well/moderate/poor	3 (11.5%)/20 (76.9%)/3 (11.5%)	2 (7.7%)/23 (88.5%)/1 (3.8%)	0.49
Cytological grade	G1/G2/G3	2 (8.7%)/13 (56.5%)/8 (34.8%)	2 (8.0%)/19 (76.0%)/4 (16.0%)	0.30
Tumor capsule formation	Well capsulated	6 (23.1%)	6 (23.1%)	0.60
	Partially capsulated	12 (46.2%)	15 (57.7%)	
	Non-capsulated	8 (30.8%)	5 (19.2%)	
Invasion of tumor capsule	No vs. Yes	14 (53.8%) vs. 12 (46.2%)	18 (69.2%) vs. 8 (30.8%)	0.25
Satellite nodule	Absent vs. Presented	23 (88.5%) vs. 3 (11.5%)	25 (96.2%) vs. 1 (3.8%)	0.30
Safety margin	>10 mm vs. <10 mm	8 (30.8%) vs. 18 (69.2%)	8 (30.8%) vs. 18 (69.2%)	1.00
Microvascular invasion	Absent vs. Presented	19 (73.1%) vs. 7 (26.9%)	20 (76.9%) vs. 6 (23.1%)	0.75
Cirrhosis	Absent vs. Presented	17 (65.4%) vs. 9 (34.6%)	13 (50.0%) vs. 13 (50.0%)	0.26
ALT (U/L)		26 (10–162)	54 (10–180)	0.10
AST (U/L)		34 (14–184)	46 (22–87)	0.92
Platelet ((10^3^/μL)		168 (96–400)	201 (97–311)	0.24
Alb (g/dL)		4.5 (3.4–5.1)	4.4 (3.5–5.3)	0.77
Bil-T (mg/dL)		0.7 (0.3–4.3)	0.7 (0.2–2.7)	0.67
PT (INR)		1.0 (0.9–1.2)	1.0 (0.9–1.1)	0.16
Ascites	Absent vs. Presented	25 (96.2%) vs. 1 (3.8%)	24 (96.0%) vs. 1 (4.0%)	0.98
CTP score	5/6/7	22 (88.0%)/2 (8.0%)/1 (4.0%)	22 (91.7%)/2 (8.3%)/0 (0.0%)	0.61
ALBI score		−3.18 (−3.67~−2.18)	−3.09 (−3.95~−1.88)	0.87
Baseline AFP before surgery (ng/mL)		8.5 (1.1–60,500.0)	251.6 (0.9–3076.0)	0.34
APRI score		0.57 (0.18–1.93)	0.65 (0.22–1.92)	0.98

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
