# Peer review of "Higher Risk of Tumor Recurrence in NASH-Related Hepatocellular Carcinoma Following Curative Resection"

_viruses, 2022, doi:10.3390/v14112427_

Round 1

Reviewer 1 Report

The manuscript is well written. There are some minor issues to be addressed.

1. The primary outcomes were OS, TTR and RFS (line 19). However only TTR was mentioned in the abstract. The data of OS and RFS should be summarized in the abstract section.

2. The analysis method of RFS, a time-to-event outcome, was missing in the statistical analysis section (line 93-107). In the paragraph of line 159-164, the data of RFS were presented as RFS rates at certain landmark time points. In the paragraph of line 205-208, RFS was analyzed with Kaplan-Meier analysis. Please clarify.

3. In line 161, the sentence 'a significantly shorter time to RFS' is confusing. It should be 'a significantly shorter RFS'.

Reviewer 2 Report

The manustript showed the different time of earlier tumor recurrence between patients with NASH and virus related hepatocellular carcinoma. I have following question about that.

1. How did you defined "earlier recurrence", please list the original reference.

2. The statitical methods should be more detailed and cite some classical papers.

3.Would you like to provide the reproducibele data and methods.
